# A Risk–Benefit Analysis of First Nation’s Traditional Smoked Fish Processing

**DOI:** 10.3390/foods12010111

**Published:** 2022-12-26

**Authors:** David D. Kitts, Anubhav Pratap-Singh, Anika Singh, Xiumin Chen, Siyun Wang

**Affiliations:** 1Food Science, Food Nutrition and Health, Faculty of Land and Food Systems, The University of British Columbia, Vancouver, BC V6T 1Z4, Canada; 2NHP Research Group, Centre for Applied Research and Innovation (CARI), British Columbia Institute of Technology, Burnaby, BC V5G 458, Canada; 3School of Food and Biological Engineering, Jiangsu University, Zhenjiang 212013, China

**Keywords:** First Nation’s smoke processing, safety, nutritional quality, fish, lipid oxidation

## Abstract

First Nations (FN) communities have traditionally used smoke to preserve fish for food security purposes. In this study, an assessment of chemical and microbiological food safety, together with nutritional quality, was conducted on fish preserved using traditional smoke processing. High-molecular-weight polycyclic aromatic hydrocarbons (PAH) residues accounted for only 0.6% of the total PAH in traditionally fully smoked salmon, and Benzo(a)pyrene (B(a)P) was not detected in the FN smoked or commercial smoked fish, respectively. The antimicrobial activity of the solvent extracts derived from smoked fish towards *Listeria innocua* was very low but detectable. The practice of using full and half-smoked processing for fish reduced all of the fatty acid concentrations and also minimized the further loss of essential omega-3 fatty acids to a greater extent than non-smoked fish during storage (*p* < 0.05). This finding corresponded to lower (*p* < 0.05) lipid oxidation in smoked fish. We conclude that the benefits of reducing lipid oxidation and retaining essential fatty acids during storage, together with a potentially significant reduction in *Listeria* contamination, are notable benefits of traditional smoke processing. Although B(a)P was not detected in FN smoked fish, attention should be given to controlling the temperature and smoking period applied during this processing to minimize potential long-term risks associated with PAH exposure.

## 1. Introduction

Fish and fish products represent an important food staple in the diet of First Nation communities that live in rural Northern Canada. The use of smoke processing for fish and seafood continues to be practiced as a traditional method of preservation in North American First Nation communities [1,2]. Smokehouses typically use hot smoke generated by burning wood on an open fire to directly expose fish to both smoke and heat over varying time periods. In British Columbia, Canada, First Nation communities categorize smoked fish products based on the length of the smoking time; with partially smoked fish requiring 2–3 days compared to fully smoked fish, which can take up to 5–6 days [2]. These products are considered ready-to-eat (RTE) and are used in a manner similar to commercial smoked RTE fish that has been processed using low heat.

The traditional smoke processing used by First Nations communities involves direct smoking, whereby fish are placed above a heat source (e.g., a wood source, such as pine and birch) to generate smoke without controlling for the heat intensity or smoke exposure [2,3]. The potential concerns of using this hot smoke process include the combination of high temperature and the inherent unique lipid compositional properties of the fish that can potentially produce undesirable by-products, such as polycyclic aromatic hydrocarbons (PAHs) [4,5,6]. The direct pyrolysis of fat melting onto the heat source, together with the smoke produced from the incomplete combustion of the wood used to generate heat, collectively produces PAHs that accumulate mostly on the outside of the fish [7]. Symptoms of short-term exposure to PAHs can include skin and eye irritation and immune, reproductive, and developmental toxicities [8]. In contrast, long-term exposure to fifteen different PAHs, of which eight are high-molecular-weight isoforms, has raised concerns of genotoxicity, mutagenicity and carcinogenesis [9,10,11]. Regulatory bodies have designated Benzo(a)pyrene (B(a)P) as a marker of the occurrence of carcinogenic PAH in food in order to evaluate the PAH contamination of commodities, whereas other PAHs that are not identified as carcinogens may act as synergists [12]. B(a)P residues have been reported to exceed the set limits of 5 μg/kg [9] when processed with different durations of smoking and using different wood sources [13]. In our previous study, we detected only trace amounts (<5 μg/kg) of B(a)P in fully smoked fish [2]. 

Another concern related to RTE fish products is the risk posed by exposure to foodborne pathogens, mainly *Listeria monocytogenes* [14]. It is noteworthy that raw salmon is not so much the source of *L. monocytogenes* compared to the significance of the processing environment and hygienic practices that result in the recontamination of the product with *Listeria* sp. [15]. We reported that the preservation methods used by the First Nations showed no signs of *L. monocytogenes***,** albeit the presence of microorganisms that signaled breaks in hygiene was noted [16]. In a former study, we also showed that the extracts derived from smoked fish were effective at controlling the growth of *Staphylococcus aureus* [17]. 

Salmon is a major food source harvested by First Nations communities in B.C. The nutritional benefits associated with salmon consumption include its high-quality protein content and that it is also an excellent source of highly polyunsaturated fatty acids (HUFAS), especially docosahexaenoic (DHA) and eicosapentaenoic (EPA) acids [18,19]. These two important HUFAs have been attributed to numerous health benefits that include preventing cardiac arrhythmia (ventricular tachycardia and fibrillation) and sudden death, as well as anti-thrombotic functions, which is a known factor linked to reducing myocardial infarction [20,21]. Moreover, EPA and DHA have well-recognized cognitive benefits that include improving attention deficit hyperactivity disorder, dyslexia, and depression and have a positive contributing effect on child IQ scores at 4 years of age [22,23]. However, after catching fish, the susceptibility of HUFAS to lipid oxidation can represent a potential food safety and health risk issue that involves the hydrolytic degradation and autooxidation of HUFA during storage, which not only lowers the nutritional value of the fish products but is also a source of unwanted lipid oxidation products [24,25,26,27]. Thus, reducing lipid oxidation reactions through effective processing is vital to retaining the safety and nutritional value of the salmon consumed by First Nations communities. 

The purpose of this study was to investigate the effects of traditional FN smoke preservation methods on potential PAH formation, along with quantifying the losses in the nutritional value of the lipid content in smoked fish. By measuring both the chemical and microbiological parameters, we attempted to provide a more descriptive risk–benefit analysis of using traditional smoke processing to preserve fish than previously reported.

## 2. Materials and Methods

### 2.1. Sample Collection and Preparation

Fish, in this case, salmon, were collected by four First Nation communities in Northern B.C. and smoked in similar smokehouses using the protocols reported in our earlier studies [2,16]. Briefly, prior to smoking, all of the fish were fileted and salted. The treatments consisted of fish that were half-smoked (*n* = 6) for later use as canned products and fully smoked (*n* = 6), which no longer required further processing. The non-smoked control fish (*n* = 6) were also obtained from the FN communities. In addition, three commercial-smoked salmon samples (Finest at Sea Ocean Products Ltd., Vancouver, BC, Canada) were purchased to be included for the qualitative comparisons only. The actual time periods used to smoke the fish, thus distinguishing between the partially and fully smoked products, were the same as described previously [2], this being 2–3 days and 5–6 days, respectively. The samples of smoked fish were stored in re-sealable Ziploc bags on ice and delivered to the Food, Nutrition and Health Building, located on the Point Grey University of British Columbia campus, Vancouver, British Columbia. The samples were obtained within 48 h of shipping. Upon receipt of the samples, the fish were thawed and homogenized using a commercial blender (Waring, Stamford, CT, USA) and then stored at −20 °C until ready for further analyses.

### 2.2. Moisture Measurement

The moisture content of the fish samples was determined according to published methods [28]. Each treatment was performed in triplicate.

### 2.3. Sodium and Potassium Analysis

The samples were measured for sodium and potassium using ICP-MS (Maxxam Analytics, Burnaby, BC, Canada) and expressed on a dry-weight basis [16].

### 2.4. Polyaromatic Hydrocarbon (PAH) Analysis

The PAH analysis of the homogenized samples was conducted by Maxxam Analytics (Burnaby, BC, Canada). The homogenized fish (5 g freeze dried) and a methanol blank were spiked with a surrogate standard mixture containing 4 deuterated PAHs. The samples were extracted with 100% dichloromethane (DCM) at 100 °C at a pressure of 2000 psi (Dionex—model 2000). The DCM extracts were passed through a sodium sulfate column to remove residual water and then exchanged into hexane. The hexane extracts were transferred to a fully activated silica gel clean-up column and eluted with 40 mL of hexane, followed by 70 mL of 50% DCM in hexane. The polycyclic aromatic hydrocarbon was recovered in iso-octane and concentrated before adding the internal 3 pre-deuterated PAH standards. Samples were analyzed by GC-MS (Agilent 6890 gas chromatograph equipped with a Model 5973 mass selective detector). A GC capillary fused silica column (Agilent HP-5ms; a length of 30 m, an inner diameter of 0.025 mm, and a film thickness of 0.25 μm) was used. The samples were injected (2 μL) in the pulsed split-less mode with an initial pressure of 25 psi held for 1.2 min and then maintained a constant flow of 1.2 mL/min. The column temperature was programmed as follows: 80 °C (2 min), 80–100 °C (50 °C/min), 100–300 °C (5 °C/min), and 300 °C (5 min). The injector temperature was set to 260 °C, the transfer line to 300 °C, and the ion source and quadrupole to 230 °C and 106 °C, respectively. Helium was used as the carrier gas. Two ions were monitored for each analyte and per-deuterated PAH standards. The PAH content was determined based on a 5-point calibration curve that was checked after every 6 injections for continuing performance. Authentic standards were used to generate the relative response factor for the PAH. The detection limits and percent recovery of individual PAHs varied with specific compounds but was less than 1 ng/g, with recoveries ranging from 70–85% [2]. To obtain an estimate of the toxic potency of the PAHs in the smoked fish samples, we calculated a total PAH benzo(a)pyrene equivalent according to the equation [29]: BaP eqi = ∑ (BaP eqi) = ∑ (C_PAH_i × TEF_PAHi_)(1)
where BaP eqi value represents each PAH from the concentration samples (C) multiplied by the toxic equivalency factor (TEF), derived from Appendix A [30].

### 2.5. Fatty Acid Analysis

The total crude lipids were recovered from the fish extracted in duplicate using Folch’s method consisting of chloroform: methanol (2:1 *v*/*v*), containing 0.01% butylated hydroxytoluene (BHT) as the antioxidant [31]. Fatty acid methyl esters (FAME) were prepared from the total crude lipids by transesterification with Boron trifluoride (BF_3_) in methanol [32]. An internal standard (tricosanoic acid methyl ester, C23:0, 10 mg/mL solution) was added to the sample for the quantification analyses. FAMEs were extracted in hexane and analyzed using gas chromatography (GC-17A, Shimadzu Scientific Instruments Inc., Columbia, MD, USA) equipped with a flame ionization detector and an auto-injector (AOC1400, Shimadzu Scientific Instruments Inc., Columbia, Maryland). The samples were injected into a capillary column (30 m × 0.25 mm; 0.25 µm film thickness; liquid phase: J&W DB 23) with helium as the carrier gas. The column temperature was initially set to 153 °C for 2 min, then increased to 174 °C for 2.3 °C/min, and to a final temperature of 220 °C for 2 °C/min. The detector and injector temperatures were both set at 250 °C. The chromatographic peaks of the fatty acids were integrated and identified using the Shimadzu software package (version 7.2.1 SP1) and were then compared to known fatty acid standards (GLC 463 and GLC 68B), as supplied by Nu-Chek Prep, Inc. (Elysian, MN, USA) and a sample of well-characterized Menhaden fish oil (PUFA-3 from Matreya, Inc., Pleasant Gap, PA, USA). The individual fatty acids were calculated and reported as weight percent of the total identified fatty acids using the peak area percentages and mass response factors relative to C18:0. The contents of the most nutritionally important fatty acids, EPA and DHA, were additionally analyzed using the internal standard C23:0 FAME.

### 2.6. Lipid Oxidation Analysis

The fish samples (30 g) were placed in sealed tubes under a normal air headspace and stored at refrigeration temperature for 30 days to record the generation of oxidation products. The lipid peroxidation of the fish lipids was assessed using the FOX assay for primary lipid oxidation products, hydroperoxides [33], and malonaldehyde [34], a secondary lipid peroxidation product. To calibrate the lipid peroxides, a hydrogen peroxide solution was used to oxidize ferrous to ferric ions and to generate a linear standard curve (0–120 µM). One gram of ground fish was homogenized in 4 mL propanol and added to the FOX reagent (xylenol orange) along with ferrous ammonium sodium sulfate. Absorbance readings were taken at 560 nm using a Shimadzu UV spectrophotometer (Columbia, MD, USA) and converted to lipid peroxides using the equation: Lipid peroxides = Abs 560 _nm_ × 133.5 − 1.765(2)
where Lipid peroxides (µM); Abs 560 _nm_ and slope of standard curve = 133.5.

The MDA content of the fish was from 5 g filet tissue blended in a homogenizer with distilled water containing trichloroacetic acid (7.5% *w*/*v*), propyl gallate (0.1% *w*/*v*, Sigma), and ethyenediaminetetracetic acid (0.1% *w*/*v*). The filtrate was reacted with 2-TBA (0.02 M, Sigma) and heated in a boiling water bath for 40 min. Absorbance was recorded at 532 nm using a Shimadzu UV spectrophotometer with a standard curve constructed using MDA standard (1,1-3-3-tetrathoxypropane (TEP, Sigma). The malonaldehyde content of the samples was expressed in units of mg MDA/100 g tissue.

### 2.7. L. innocua Inhibition Assay

The fish extracts were obtained using a sequential hexane extraction (e.g., 1:5 *w*/*v*; 3×) followed by distilled water extraction in triplicate. The hexane extracts were filtered and evaporated under a vacuum at 45 °C. The water extracts were freeze-dried. The dried extracts were weighed to determine their yield. All of the extracts were frozen (−20 °C) until used.

We used *Listeria innocua* (FSL C2-008) to model microbial growth. The bacterial strain was isolated from a fish processing plant, and the stock was kept frozen at −80 °C before use. The inoculum was thawed and streaked onto Brain–Heart Infusion (BHI) agar using a sterile disposable loop before incubation at 37 °C for 24 ± 1 h. A colony of *L. innocua* was identified from the overnight culture, recovered, and washed three times in PBS (Difco) and centrifuged (1500 RPM for 5 min, 20 °C). The pellets were re-suspended in PBS to contain initial bacteria concentrations of 10^9^ cfu/mL. Serial dilutions were made in PBS to achieve a starting concentration of approximately ca 10^6^ cfu/mL. Fresh cultures prepared in 5 mL of BHI broth were prepared for each experiment and the incubation period was standardized for 18 h at 37 °C. 

To generate the *Listeria innocua* growth curves, 5 mg of the hexane or water extract collected from different fish samples (e.g., commercial smoked, FN non-smoked, FN half-smoked, FN fully smoked) were dissolved in dimethyl sulfoxide (DMSO, Thermos Scientific^TM^, Waltham, MA, USA) and BHI broth. Aliquots (100 μL) of the diluted stock culture were added to each of the extraction samples to achieve an initial bacterial concentration of ca 10^5^ cfu/mL. The control tubes were devoid of fish extracts. An initial aliquot was withdrawn immediately to establish a time zero reading, with the remaining transferred to a shaking incubator set to 37 °C. Subsequent aliquots were withdrawn at 2 h intervals over the total 8 h test period. Appropriate dilutions were made in PBS before surface plating. The experiments were conducted in triplicate, with three rounds of sampling. For the first round, three dilutions for each treatment at each time point were plated; while for the second and third rounds, two dilutions were performed in order to yield countable numbers. The BHI plates were incubated at 37 °C for 24 h. The experiment was conducted three times in total to establish *L. innocua* growth curves. A new single colony was selected every time to prepare the overnight culture. The lag times used to define the relative inhibitory effects of the different fish extracts on *L. Innocua* were determined using the Gompertz equation [35].

### 2.8. Statistical Analysis

All of the results are presented as means ± Standard Deviation (SD). Statistical analysis was conducted using One-Way or Two-Way Analysis of Variance with Tukey’s post hoc tests (MINITAB software (Version 14, Minitab Inc., State College, PA, USA) with *p* < 0.05 representing a statistically significant difference.

## 3. Results

The moisture content of FN non-smoked salmon (72.4 ± 1.79%) was higher (*p* < 0.05) than FN half-smoked (65.5 ± 3.01%) and markedly higher (*p* < 0.001) than the FN fully smoked (10.7 ± 0.8%). Comparatively, the moisture content of commercial smoked fish (67.0 ± 4.2%) resembled that of the FN half-smoked product. The corresponding sodium and potassium concentrations expressed in the dry weight of the fish were considerably higher both in the FN half-smoked (Na-13.5 ± 3 ppm; K-16.1 ± 2 ppm) (*p* < 0.05) and FN fully smoked (Na- 81 ± 9 ppm; K-103 ± 11 ppm) fish compared to the non-smoked control (Na-0.9 ± 0.1 ppm; K-5.9 ± 0.8 ppm). We attribute these differences to both the practice of applying salt to fish prior to smoking as a pre-curing process as well as the degree of dehydration that resulted from the duration of the smoke processing. The total fat content (g/100 g) of FN non-smoked salmon (17.86 ± 1.67) was greater (*p* < 0.05) than both FN half-smoked (12.95 ± 1.14) and FN fully smoked (12.24 ± 1.3). For the sake of comparison, the fat content recovered from the commercial smoked salmon reference was lower (11 ± 2.0).

The PAH profile of the FN smoke-processed fish is expressed on a dry weight basis and is presented in Table 1. There were no detectable PAHs recovered from the non-smoked FN or commercial smoked fish controls. In the FN smoked fish, however, a number of low-molecular-weight PAHs were identified and quantified in both the half-smoked and fully smoked fish samples. The concentration of PAHs recovered from the FN half-smoked products was only 2.5% of the total PAHs recovered from the fully smoked fish samples (*p* < 0.01). Of the 13 different PAHs, the concentrations of low-molecular-weight PAHs (e.g., 2–4 aromatic rings, including naphthalene, acenaphthylene, acenaphthene, fluorene, phenanthrene, anthracene, fluoranthene and pyrene) were present in only small concentrations compared to the fully smoked fish. Phenanthrene was the most dominant low-molecular-weight PAH in both FN half- and fully smoked fish samples. In addition, the fully smoked fish also contained a few high-molecular-weight PAHs (e.g., 5–6 aromatic rings), namely benzo(a)anthracene, (B(k)P) and chrysene. Other high-molecular-weight PAHs, for example, benzo(b)fluoranthene (B(b)F), benzo(k)fluoranthene, (B(f)F and benzo(a)pyrene (B(a)P were not detected in FN half-smoked or fully smoked fish, respectively. The yield of ∑ (BaP eqi), a product of the concentration of individual PAH and the toxic equivalent factor, was approximately 350 times greater in the FN smoked samples, which reflected the difference in total PAH recovered between FN half-smoked and FN fully smoked fish.

The individual fatty acid concentrations present in the FN smoked and non-smoked fish are presented in Appendix A. A summary of the nutritionally important fatty acid groups is shown in Table 2. In general, all fatty acids, regardless of the degree of saturation, were lower (*p* < 0.05) in fish after undergoing the thermal half-smoked and fully smoked processes, respectively. Of interest was the observation that very little difference occurred in the individual fatty acids between the fish that were half-smoked or fully smoked. A notable reduction in fatty acids was also observed in the heat-treated commercial smoked fish, which for the purpose of this study was used only as a reference point, as the starting fish source was different from that used by FN; notwithstanding as well the smoking procedure, which was also very different. Fatty acids were likely lost quickly during the initial heat process used with smoking, a response of fat liquefying at the high temperatures used and the quality of the fat characteristic of salmon. The relative loss of polyunsaturated fatty acids was greater (*p* < 0.05) compared to saturated fatty acids, which we attribute again to the lower melting point of unsaturated fatty acids. For example, the relative losses in omega-3 (range- 41–45%) and omega-6, (range- 42–44%) were highest after smoking compared to saturated fatty acids (range- 27–34%). A summary of the changes in the important fatty acid subclasses attributed to smoking is presented in Table 2. The omega-6 PUFA content in thr fish was present in relatively lower concentrations compared to omega-3 fatty acids; however, both fatty acid groups lost approximately 50 percent during the specific smoke processing. The omega-3/omega-6 ratio also appeared to be higher, but this again was not significant and, in general, reflected the proportionally similar loss of lipids for all treatments. 

The application of traditional FN smoke preservation to the fish resulted in lipid oxidation (*p* < 0.05), which was monitored over a 30-day period (Figure 1). This observation was particularly striking in the processed fish that received the full-smoke protocol. The temporal patterns of lipid hydroperoxide generation at refrigeration temperature were fairly constant in all of the processing treatments, with the exception of the FN fully smoked fish; characterized as having a lag phase of approximately 8 days before propagation occurred (Figure 1A). For comparison purposes, the same trend was also observed for commercial salmon that had not been smoked.

In the non-smoked fish, the primary products of lipid oxidation appeared rapidly up to day 21, reaching levels that were higher *p* < 0.05) than the values observed in the half-smoked fish. The onset of hydroperoxides production in the FN fully smoked fish remained very low and significantly lower than that obtained in the FN half-smoked fish (*p* < 0.05). A similar trend was observed for the secondary products of lipid oxidation, namely malonaldehyde monitored at storage days of 9, 21, and 30, respectively (Figure 1B). 

The growth curves of *Listeria innocua* over 8 h obtained in the presence of water and hexane extracts of non-smoked and FN smoked fish are shown in Figure 2. *Listeria innocua* growth was not mitigated by exposure to the water extract (Figure 2A) but did show significant (*p* < 0.05) inhibition to the hexane extracts (Figure 2B) collected from only FN fully smoked salmon samples. The time defining a lag phase for *L. innocua* growth was approximately 3 h, 4 h, and >8 h, respectively, for commercial smoked, half-smoked, and fully smoked hexane extracts. This corresponded to 19.4 ± 3.55 and 23.4 ± 2.20 percent of growth inhibition for the hexane extracts derived from the commercial smoked and FN half-smoked salmon, respectively. For interest, the hexane extracts collected from the fully smoked salmon samples produced the highest suppression (34.2 ± 0.5%), which was attributed to the prolonged lag phase.

## 4. Discussion

All of the smoked salmon samples gifted to us by Canada’s First Nation communities of Tl’azt’en and Lheidli T’enneh were obtained from a traditional smokehouse that employed procedures that did not control temperature and time. A commercial smoked salmon, which was obtained from a different source, was included in the analysis for the purpose of having a commercial reference. The smoking times for the FN half-smoked and FN fully smoked samples were typically 2–3 days and 5–6 days, respectively, which was identical to our previous study [2], but in comparison, much longer than that used by others [36,37]. The uncontrolled smoking temperature and the long smoking time used in their traditional method of smoke processing are two important factors related to generating both the level and type of PAHs reported in this study. 

Curing fish using cold or hot smoke methods ultimately reduces the moisture content and, most importantly, water activity. Thus, the lower moisture contents recorded in both of the FN smoked fish products relative to a commercial product and our FN non-smoked product were expected. This change corresponded to considerably higher salt contents, most notably measured in the FN fully smoked products. Processing fish at high temperatures without controlling the distance from the heat source is common to the traditional FN smoke processing employed herein and is thus a critical factor in explaining the final moisture results of the smoke-processed fish. For smoking to be effective for drying, sufficient time is needed for the heat to penetrate the muscles of the fish, resulting in the release of bound water and the substitution of moisture with the concomitant uptake of volatile organics derived from the wood heat source. Once the internal temperature is reached, the drying process and the moisture content will be affected by the duration of smoking. This was observed in the present study with the FN fully smoked fish exposed for up to 2 to 3 days longer than the FN half-smoked product using the same traditional process. The dehydration process, therefore, can be described as having sufficient time for heat derived from the wood fire to penetrate into the muscles of the fish, reaching an internal temperature that facilitates a high rate of moisture loss over time. The use of salt as a pre-application step, along with the simultaneous uptake and retention of smoke volatiles [38], would produce low water activity and low-moisture final dried product, which in combination, preserves the quality (e.g., color, texture) [39]. The high salt content, despite having important preservation qualities, also presents potential health risks if the smoked fish cured using this method is consumed regularly [16].

The presence and concentrations of individual PAHs reported by others in smoked salmon are highly variable due to differences in the smoking processes, in particular, the smoke generation conditions that include the temperature reached for pyrolysis and the smoking duration, the fats lost due to liquefaction during heat exposure, and the type of wood used to smoke-process the food products [40]. In the present study, the PAH contents of the smoked fish were only derived during the smoking process, as no PAHs were detected in the FN non-smoked fish or the commercial salmon samples, thus ruling out environmental contamination as a cause for PAH residues. The total PAH content in our smoked fish was similar to the levels reported in an earlier study [2] and was similar to those by other investigators [13,41]. We confirmed the observation that greater amounts of total PAHs were recovered from the FN fully smoked salmon compared to the FN half-smoked salmon, thus identifying the smoking time used in traditional FN preservation methods as a primary factor influencing the total PAHs recovered from the FN smoked salmon. We used safety standards designed for PAHs relative to B(a)P (e.g., Toxic equivalents of B(a)P)) to show that the PAHs present in the fully smoked salmon were markedly greater than that present in the half-smoked salmon.

An important finding was that low-molecular-weight PAHs predominated in FN smoked salmon, which is similar to our previous results [2]. Of these particular PAHs recovered, fluorene, phenathrene, napththalene, benzo[a}anthracene, and chrysene have a Group 2B classification as possible human carcinogens [42,43]. Naphthalene, the most concentrated PAH recovered in both the FN half-smoked and fully smoked fish poses potential health risks associated with chronic exposure that includes both cataracts and retinal hemorrhage, whereas headache, nausea, and anemia can occur with acute exposure (44). Occupational workers exposed to naphthalene have recorded laryngeal carcinomas or neoplasms in the pylorus and cecum [44]. Chrysene was only recovered from the FN fully smoked fish, which confirmed our earlier reports [2]. Human data on chrysene carcinogenicity are unavailable; however, it has been reported to induce carcinomas and malignant lymphoma in mice [45]. Chronic exposure to chrysene has also been associated with disorders that include immunologic deficiency syndromes and kidney, liver, and lung neoplasm [45]. 

Unlike our previous study, higher molecular weight PAHs that included B(a)P were not detected herein in both FN half-smoked and fully smoked fish, respectively. The fact that we did not reproduce our previous finding in this respect is of interest and could be related to the differences in the wood sources used between the studies. For example, available wood sources during our two studies included mixtures of spruce and alder. The relative amount of each wood source used in our former study was not recorded; however, in the present study, we requested that spruce be preferentially used, which is a relatively soft wood and lower than many other kinds of wood. Others have reported a higher risk associated with carcinogenic and mutagenic toxicity equivalents occurred when hardwoods were used to smoke-cure fish for longer durations [13]. Legislation for the maximum levels of B(a)P for smoked meats and smoked meat products set in 2006 is 5 μg/kg, but there are many cases where B(a)P exceeded this. Japanese workers, for example, reported that smoked, dried fish products derived from bonito contained high levels of B(a)P residue [46], which more recently was reduced by countermeasures using smoking on a hot plate with wood chips [47]. B(a)P, classified as a Group 1 carcinogen, was chosen as a marker for potential carcinogenic risks of PAH exposure in food. More recent evidence from a number of in vivo bioassays using experimental animals have identified up to 15 PAHs (e.g., benz[a]anthracene, benzo[b]fluoranthene, benz[j]-fluoranthene, benzo[k]fluoranthene, benzo[ghi]perylene, benzo[a]pyrene, chrysene, cyclopenta[cd]pyrene, dibenz[a,h]anthracene, dibenzo[a,e]pyrene, dibenzo[a,h]pyrene, dibenzo[a,i]pyrene, dibenzo[a,l]pyrene, indeno[1,2,3-cd]pyrene, and 5-methylchrysene) with potential adverse health risk effects from long-term dietary intake [48]. Of the 15 PAHs identified above, only small amounts of chrysene and benz[a]anthracene were only detected in the FN fully smoked products. Thus, addressing the potential carcinogenic potential of exposure to the PAH mixtures identified in this study is muted by the fact that PAHs with chronic health concerns were either not present in FN smoked fish or only occurred in trace amounts in the fully smoked fish products. Nevertheless, future studies addressing how to optimize the smoke-processing to eliminate all PAHs are warranted. This goal would follow the Codex Alimenarius Commission, the code of practice, which called for reducing the PAH contamination of food processed using smoke curing [49].

The application of traditional FN smoke processing methods for fish resulted in significant losses in all classes of free fatty acids compared to non-smoked control (Appendix A and Table 2). This observation corresponded to the proportional loss of approximately 27% total fat in FN smoked fish relative to the non-smoked control. Others have also reported losses in total fish PUFA following smoke processing [50]. Despite the proportional losses in MUFA, PUFA, and SFA, respectively, the FN smoke processing protocol did not affect the high omega-3 fatty acids/omega 6 ratio that typically exists in salmon and reflects their cold-water habitat. In wild Atlantic salmon, an n-3 to n-6 PUFA ratio of 11 was compared to a ratio of 3.6 in farmed Atlantic Salmon [51]. Using the salmon source in this study, our respective n-3 to n-6 PUFA ratios ranged from approximately 6 to 7, with no appreciable change attributed to the different FN smoking procedures. Although there was a 50 percent loss in both EPA and DHA due to the smoke processing, these fish were still sources of HUFA-n-3, which suggests that increasing daily fish intakes could accomplish reaching daily recommendations of total EPA + DHA, proposed for health benefits [52].

Parallel to these findings was the fact that a 20–30-day cold-storage trial for fish produced characteristic patterns of shelf-life, depicted by a relative tendency to generate lipid hydroperoxide (primary) and malonaldehyde (secondary) lipid oxidation products in the non-smoked fish. It is expected that the proportionally greater n-3 PUFA content in fish tissues enhances the relatively greater susceptibility for lipid peroxidation [53], with MDA representing the principal secondary peroxidation product of n-3 PUFAs, compared to n-6 PUFA [54]. Moreover, exposing fish to heat during smoke processing results in triglyceride and phospholipids hydrolysis and the subsequent yield of free fatty acids that are available for autooxidation during subsequent storage, if not lost during smoking. Related to the stability of PUFAs in smoke-processed fish was the observed slower rate of lipid peroxidation when fish received the FN smoke procedures. Normally, smoked, refrigerated fish products are shelf stable with a shelf life of 21 to 35 days, depending on the category of smoking applied [55]. In this study, the products of primary oxidation, namely hydroperoxides, occurred rapidly in non-treated salmon but were reduced in the FN half-smoked and especially markedly lower in the FN fully smoked salmon samples. Secondary by-products of lipid oxidation, including some off-flavor compounds and malonaldehyde, are generated from hydroperoxides, and typically, malonaldehyde levels that exceed 2 mg MDA/kg are regarded as producing objectionable odor and color. FN smoked fish that were stored at refrigeration temperature over prolonged periods reaching 30 days had an MDA content that was well below this threshold concentration. Hence, an anti-peroxidation effect of the FN smoke processing method was evident and can be explained by both the brine that was applied prior to smoke processing and the generation of phenolic compounds that occurred during smoking that impregnate the fish flesh and contribute to antioxidant activity [56]. Moreover, it is also likely that the peroxide content present in the fully smoked salmon samples was relatively stable, thus reducing the subsequent generation of secondary lipid oxidation products [33].

The sequential extraction of the smoked fish samples, first with water and followed by hexane, allowed us to demonstrate that hydrophobic compounds recovered in extracts from the FN smoked fish were responsible for the antimicrobial effect against *L. innocua*. Our former study also reported a similar antimicrobial property of hexane extracts from smoke-processed fish that corresponded to inhibiting growth of *Staphylococcus aureus* [17]. Wood phenolic compounds derived from multiple species have antimicrobial properties [57], and this has also been reported with wood smoke tested against both spoilage and pathogenic microorganisms [58]. The reasons for choosing *Listeria* as a test organism in this study were based on reports that smoked salmon is regarded as a risk factor for human listeriosis since smoked salmon can sporadically be re-contaminated with *Listeria* if systemic hygienic measures are not in place [16]. This occurs when *Listeria*, although damaged during exposure to optimal time and temperature combinations of smoke processing, proceeds to recover during extended storage [15].

## 5. Conclusions

In this study, we further explored the risks and potential benefits of traditional FN smoke-processing procedures for salmon preservation currently in use by two communities in northern B.C., Canada. According to the findings, the highest total B(a)P equivalent was ascribed to the duration of the smoke curing process in the smokehouse and involved mainly low-molecular-weight (LMW) PAHs. In accordance with these findings, B(a)P or other notable high-molecular-weight (HMW) PAHs with carcinogenic potential were not detected at levels of concern in both smoked products. Despite the fact that the smoke processing of salmon resulted in a significant loss of lipids, this effect was quantitatively similar for all fatty acid groups and, more importantly, did not adversely affect the ratio of essential n-3/n-6 PUFA. Losses of PUFA are expected with the fat type of salmon and the distance from the cooking flame during smoking. It was also important to observe that the generation of both primary and secondary lipid oxidation products was mitigated during the subsequent storage of the smoked fish. Furthermore, an apparent antimicrobial effect of FN smoke-processing, specific to the lipophilic constituents recovered from the smoked fish products, occurred with *Listeria* sp. Thus, the retention of lipid quality of smoked salmon during storage and prevention of recontamination by *Listeria* organisms during storage would be regarded as benefits of the smoke processing conditions used by FN. On the whole, FN fully smoked processing of salmon may pose a potential risk to human health since it generated the highest number of PAHs, which could, both individually or synergistically, induce non-cancer or cancer processes. However, the extent of retention of n-3/n-6 essential acids and mitigation of lipid oxidation in a stored product, not to mention the prevention of *Listeria* sp., are benefits to consider in establishing a risk assessment of FN smoke processing of fish.

## Figures and Tables

**Figure 1 foods-12-00111-f001:**
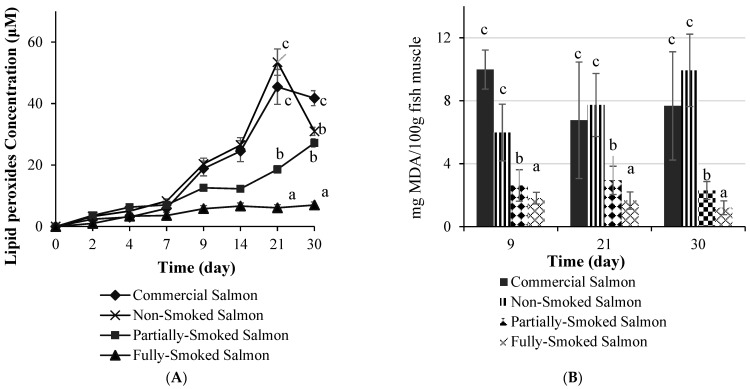
(**A**) Lipid hydroperoxide content in commercial and FN non-smoked and FN half-smoked and fully smoked FN salmon versus storage time (days); (**B**) Malonaldehyde (MDA) content in commercial and FN non-smoked salmon and FN half-smoked and fully smoked FN salmon during storage time (days). Values represent mean ± SD (*n* = 3). Superscripts with different alphabets (a–c) are significantly different (*p* < 0.05).

**Figure 2 foods-12-00111-f002:**
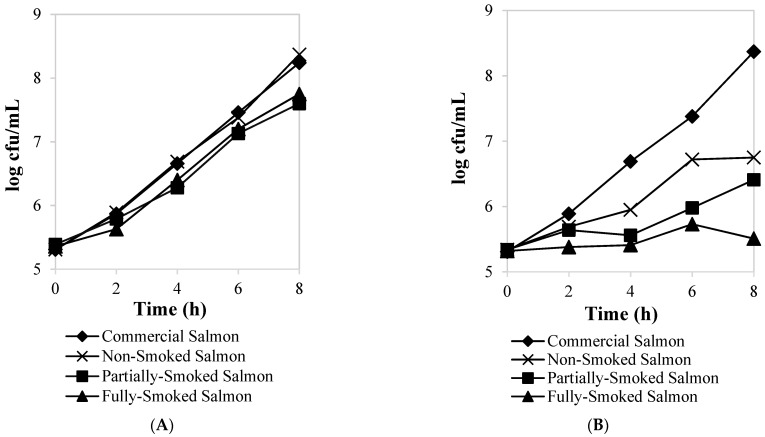
Effect of smoked salmon water (**A**) and hexane (**B**) extracts against *Listeria innocua*. Values represent means of 3 triplicates, but the SD is too small to show.

**Table 1 foods-12-00111-t001:** Polycyclic aromatic hydrocarbon (PAH) content recovered from non-smoked FN and commercial salmon and half-smoked and fully smoked FN salmon ^1^.

PAH	Non-Smoked ^2^	FN Half-Smoked	FN Fully Smoked
Naphthalene	ND ^3^	229 ± 76 ^a^	351 ± 87 ^b^
Acenaphthylene	ND	351 ± 92 ^a^	586 ± 210 ^b^
Acenaphthene	ND	31 ± 15 ^a^	117 ± 18 ^b^
Fluorene	ND	122 ± 46 ^a^	532 ± 85 ^b^
Phenanthrene	ND	382 ± 137 ^a^	3191 ± 373 ^b^
Anthracene	ND	76 ± 15 ^a^	538 ± 65 ^b^
Fluoranthene	ND	46 ± 15 ^a^	308 ± 46 ^b^
Pyrene	ND	31 ± 15 ^a^	273 ± 37 ^b^
Benzo[α]anthracene	ND	ND	13 ± 4 ^b^
Chrysene	ND	ND	27 ± 12 ^b^
Benzo[b]fluoranthene	ND	ND	ND
Benzo[k]fluoranthene	ND	ND	ND
Benzo[α]pyrene	ND	ND	ND
Total PAH	ND	1145 ± 382 ^a^	5972 ± 544 ^b^
Total BAP eqi. ^4^	ND	0.0109	3.97

^1^ Values represent a dry weight (µg/kg) and are expressed as Mean ± standard deviation. Means with different letters are significantly different (*p* < 0.05). ^2^ Refers to both commercial and First Nation (FN) non-smoked samples, respectively. ^3^ ND = not detected (e.g., below the detection limit of 0.001 ug/g). No PAHs were detected in the commercial cold smoked fish samples. ^4^ ∑ (BaP eqi) = Toxic equivalent factors for total BA.

**Table 2 foods-12-00111-t002:** Summary of nutritionally important fatty acid contents of FN non-smoked, half-smoked, fully smoked and commercial salmon ^1^.

Fatty Acid	Non-Smoked	Half-Smoked	Fully Smoked	Commercial
SFA	3.56 ± 0.33 ^a^	2.33 ± 0.29 ^b^	2.58 ± 1.08 ^b^	1.89 ± 0.67
PUFA	5.18 ± 0.41 ^a^	2.88 ± 0.31 ^b^	2.89 ± 0.33 ^b^	0.06 ± 0.03
*cis*-MUFA	8.19 ± 0.80 ^a^	6.47 ± 0.77 ^b^	6.74 ± 4.29 ^b^	2.54 ± 0.40
C18:2 n-6	0.45 ± 0.09 ^a^	0.21 ± 0.02 ^b^	0.24 ± 0.18 ^b^	0.11 ± 0.00
C20:4 n-6	0.21 ± 0.06 ^a^	0.05 ± 0.00 ^b^	0.06 ± 0.04 ^b^	0.06 ± 0.02
C20:5 n-3	1.20 ± 0.09 ^a^	0.72 ± 0.07 ^b^	0.78 ± 0.60 ^b^	0.73 ± 0.09
C22:6 n-3	2.20 ± 0.17 ^a^	1.30 ± 0.12 ^b^	1.20 ± 0.70 ^b^	1.70 ± 0.25
Omega-3	4.50 ± 0.36 ^a^	2.50 ± 0.23 ^b^	2.60 ± 0.80 ^b^	2.80 ± 0.34
Omega-6	0.77 ± 0.17 ^a^	0.33 ± 0.04 ^b^	0.38 ± 0.28 ^b^	0.28 ± 0.03
Omega-3/6	5.80	7.58	6.86	9.89

^1^ Values given are means ± standard deviations (*n* = 6). Expressed as g/100 g. Means with different superscript letters ^(a,b)^ in rows are significantly different (*p* < 0.05). Statistical analysis did not include Commercial salmon as they are not from the same source as FN salmon. See Appendix A for the complete fatty acid profiles.

## Data Availability

Data are contained within the article or in Appendix A.

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
