# Peer review of "A Risk–Benefit Analysis of First Nation’s Traditional Smoked Fish Processing"

_foods, 2022, doi:10.3390/foods12010111_

Round 1

Reviewer 1 Report 22nd December 2022

Review

Foods

Manuscript ID: foods-2060351

Title: A risk-benefit analysis of First Nation’s traditional-smoked fish processing

Authors: David D. Kitts *, Anubhav Pratap-Singh, Anika Singh, Xiumin Chen, Siyun Wang

The work entitled: „A risk-benefit analysis of First Nation’s traditional-smoked fish processing” concerns the effects of First Nation (FN) traditional smoke preservation methods on the formation of polycyclic aromatic hydrocarbons (PAH), losses of the nutritional value of lipids, and microbiological safety of fishes subjected to this process. The authors proved that traditional FN smoking did not cause the presence of PAHs with carcinogenic potential at levels of concern neither in half-smoked nor fully-smoked fishes. In the presented work smoking reduced lipids and fatty acids content but did not affect the proportion n-3/n-6 PUFA in smoked fishes. Moreover, traditional smoking indicated a significant reduction in Listeria contamination in the studied products.

This work is suitable for the „Foods” journal's aims and scope and the experiment was designed in a proper way. The major part of the quoted papers are topical works, coming from the last several years but there are also many (7) self-citations of David D. Kitts (the first author of the manuscript):

-       2. Kitts, D.D.; Chen, X.M.; Broda, P. Polyaromatic hydrocarbons of smoked cured muscle foods prepared by Canadian Tl'azt'en 505 and Llheidli T'enneh First Nation communities. J. Toxicol. Environ. Health Part A 2012, 75, 1249-1252, 506 doi:10.1080/15287394.2012.709410.

-       16. Hingston, P.; Johnson, K.; Kitts, D.; Wang, S. Safety and quality of fish and game meats prepared by First Nations communities 538 in British Columbia, Canada. J. Food Prot. 2020, 83, 896-901, doi:10.4315/jfp-19-492.

-       17. Allen, K.J.; Chen, X.M.; Mesak, L.R.; Kitts, D.D. Antimicrobial activity of salmon extracts derived from traditional First Nations 540 smoke processing. J. Food Prot. 2012, 75, 1878-1882, doi:10.4315/0362-028x.Jfp-12-010.

-       18. Huynh, M.D.; Kitts, D.D. Evaluating nutritional quality of pacific fish species from fatty acid signatures. Food Chem. 2009, 114, 542 912-918, doi:10.1016/j.foodchem.2008.10.038.

-       27. Kitts, D.D. Dietary lipids and physiological function. In Bailey's Industrial Oil and Fat Products; 2020; pp. 1-33. 562 doi:10.1002/047167849X.bio105.

-       32. Kitts, D.D.; Huynh, M.D.; Hu, C.; Trites, A.W. Season variation in nutrient composition of Alaskan walleye pollock. Can. J. Zool. 572 2004, 82, 1408-1415, doi:10.1139/z04-116.

-       33. Lai, M.M.C.; Zhang, H.A.; Kitts, D.D. Ginseng prong added to broiler diets reduces lipid peroxidation in refrigerated and frozen 574 stored poultry meats. Molecules 2021, 26, 4033, doi:10.3390/molecules26134033.

The question is if it is necessary to cite all these papers to keep the value of the work.

The other uncertainties which need to be clarified:

1.      Page 1, line 7, lacks the whole address of the institution. What stands for BC abbreviation?

2.      Page 8, line 309, incorrect numbering under Figure 1 (should be Figure 1 instead Figure 2).

3.      Page 10, line 385, the sentence „Pine and Birch are relatively soft woods compared to Popular” is not clear. It seems that this sentence does not make any sense. Why do the words „Birch” and „Popular” start with capital letters?

4.      Page 11, lines 446-448 in the sentence „Moreover, it is also likely that the content of peroxides present in the fully-smoked salmon samples were relatively more stable, thus precluding secondary product production” I suggest the word „production” replace some synonym.

5.      Page 11, lines 454-456, in the sentence „Wood phenolic compounds derived from multiple species have antimicrobial properties and this has been reported with wood smoke wood tested against both spoilage and pathogenic microorganisms”, the part „with wood smoke wood tested” seems to be confusing.

6.      Page 13, line 590, incorrect use of capital letters for the names in the references.

7.       Page 9, line 326, Some statements given in this work seems to be obvious e.g. „No PAHs were present in FN non-smoked control”.

8.      The authors admitted that the conditions of smoking (time, temperature or kind of used wood) were out of control. The question is if received data might be repeatable when FN is taking under consideration.

I accept the presented manuscript after minor revision (correcting above mentioned mistakes).

Author Response

Please see the attachment, Response to Reviewer 1.

In this response, we have made all the necessary corrections identified by Reviewer 1 and also gave a response to the first question. 

Reviewer 2 Report

In this study, the authors examined the residual PAH in smoke salmons produced as Canadian First Nation's products. As a result, authors found both the half- and fully-smoked salmons contained PAH to some extent. On the other hand, the products had some beneficial effect in terms of anti-oxidant and anti-microbial effects. The manuscript was well-written, though a few revision should be needed as described below.

Major

The criteria of PAH must be important. Authors should write this information in the fifth paragraph of Discussion as well as maximum level of B(a)P (line 388-389).

Minor

1. Authors should explain the smoking time of half-smoked and fully-smoked fish in methods section (line 91-92, 95-97). It is not recommended to make readers to look for the reference No. 2.

2. Authors should explain the effects of usage of three fish species (sockeye and Coho salmons, and trout). Were there any differences among these products?

3. Authors should explain the method to measure moisture content briefly even if there was a reference (line 104).

4. line 141. "the method described by [32]". By whom?

5. The concentrations of water and hexane extracts should be described in the legend of Figure 2.

6. Line 385. Authors should check the usage of capitals (Birch and Popular).

7. Line 467. Define abbreviations of LMW and HMW

Author Response

Please see the attachement:  Response to Reviewer 2.
